# Stillbirth in term and late term gestations in Stockholm during a 20-year period, incidence and causes

Hanna Åmark[1]*, Christina Pilo[2], Ingela Hulthén Varli[3]

**1** Department of Clinical Science and Education, Unit of Obstetrics and Gynecology, Karolinska Institute, Södersjukhuset, Stockholm, Sweden, **2** Department of Obstetrics and Gynecology, Södertälje Hospital, Stockholm, Sweden, **3** Department of Women´s and Children´s Health, Karolinska Institutet, Stockholm, Sweden

\* hanna.amark@sll.se

## Abstract

### Introduction

The incidence of stillbirth has decreased marginally or remained stable during the past decades in high income countries. A recent report has shown Stockholm to have a lower incidence of stillbirth at term than other parts of Sweden. The risk of antepartum stillbirth increases in late term and postterm pregnancies which is one of the factors contributing to the current discussion regarding the optimal time of induction of labor due to postterm pregnancy.

### Material and methods

This is a cohort study based on the Stockholm Stillbirth Database which contains all cases of stillbirth from 1998-2018 in Stockholm County. All cases were reviewed systematically and the cause of death was evaluated according to the Stockholm Stillbirth Classification. Stillbirths diagnosed between gestational week (GW) 37+0 and 40+6 n = 605 were compared to stillbirths diagnosed from GW 41+0 and onwards n = 157, according to the cause of stillbirth and pregnancy and maternal characteristics. The aim was to evaluate the incidence of stillbirth over time and the incidence of stillbirth diagnosed from GW 41+0.

### Results

In Stockholm County the overall incidence of stillbirth has decreased from 4.6/1000 births during the period 1998-2004 to 3.4/1000 births during the period 2014-2018, p-value <0.001. When comparing the same time periods, the incidence of stillbirth diagnosed from GW 41+0 and onwards has decreased from 0.5/1000 births to 0.15/1000 births, p-value <0.001. Among women still pregnant at GW 41+0 the incidence of stillbirth has decreased from 1.8/1000 to 0.5/1000. When comparing stillbirths diagnosed at GW 37+0-40+6 with stillbirths diagnosed from GW 41+0 and onwards infection was a more common cause of stillbirth in the latter group.

**Citation:** Åmark H, Pilo C, Hulthén Varli I (2021) Stillbirth in term and late term gestations in Stockholm during a 20-year period, incidence and causes. PLoS ONE 16(5): e0251965. https://doi.org/10.1371/journal.pone.0251965

**Data Availability Statement:** Due to sensitive patient information, data can only be provided on request to the ethics committee (registrator@etikprovning.se).

**Funding:** This study was supported by Spädbarnsfonden and KI Research Foundation.

**Competing interests:** The authors have declared that no competing interests exist.

## Conclusion

In Stockholm County there was a decreasing incidence of stillbirth overall and in stillbirths diagnosed from 41+0 weeks of gestation and onwards during the period 1998-2018. In stillbirths diagnosed from GW 41+0 and onwards infection was a more common cause of death compared to stillbirths diagnosed between GW 37+0 and 40+6.

## Introduction

Approximately 2.6 million fetuses are stillborn every year [1]. Globally, the incidence of stillbirth is most prominent in low income parts of the world. However, stillbirth is also a public health problem in high income countries [1] and there has only been a marginal decrease during the past decades [2]. The incidence of neonatal death is decreasing faster than the incidence of stillbirth [1, 3] hence, stillbirth is the main contributor to perinatal death in high income countries [3]. In Sweden the incidence has been quite stable during the last 20 years, about 3-/1000 births, however according to a report from the National Board of Health and Welfare, stillbirth incidence differed significantly between regions in Sweden and pregnant women in Stockholm County had a lower risk compared to the rest of the country especially at term [3].

Approximately 20 percent of pregnant women have not gone into spontaneous labor at gestational week (GW) 41+0 [4]. In Sweden, postterm pregnancy is defined as a gestational age of $\geq$ 42+0 weeks. The frequency of women still pregnant at GW 42+0 was in 2018 approximately 9% among nulliparous and 5% among multiparous [5]. The risk of stillbirth increases in late term and postterm pregnancies [6]. Due to reports of increased risk of stillbirth in late term (GW 41+0-41+6) and postterm (GW $\geq$ 42+0) pregnancies the recommendations of surveillance of pregnancies and time of induction has changed in Stockholm County. In 2005 the recommended date for induction due to postterm pregnancy was changed from GW 43+0 to 42+0 in all delivery units in Stockholm County [7]. In 2014 the surveillance from GW 41+0 was standardized and extended at all delivery units [8].

A recent study comparing induction of labor at GW 41+0 to expectance and induction latest at GW 42+0 showed a lower risk of stillbirth in the former group [9]. These results and varying management policies between countries regarding postterm pregnancies have actualized the discussion regarding the optimal time of induction of labor due to prolonged pregnancy.

Fetal abnormalities, placental insufficiency, fetal growth restriction and infections are all common causes of stillbirth as is placental abruptio and umbilical cord complications [10, 11]. Main causes of stillbirth vary with gestational age [10, 12]. Pregnancies allowed to pass gestational week 41+0 are in general without risk factors and without known pregnancy complications.

The aim of this study was to investigate if the incidence of stillbirth diagnosed from 41 completed weeks and onwards as well as the overall incidence of stillbirth has changed over the past two decades in Stockholm County. And in addition, to investigate the specific causes of stillbirth among stillbirths diagnosed from GW 41+0 compared to stillbirths diagnosed between GW 37+0 and 40+6.

## Material and methods

This is a cohort study based on the Stockholm Stillbirth Database. The Stockholm Stillbirth Database contains all stillbirths in Stockholm County 1998-2015. Since 2016, stillbirth cases

are instead included in the Swedish Pregnancy Register, a certified national pregnancy register. Maternal characteristics, pregnancy complications, laboratory and microbiology findings, placental pathology, chromosomal analyses and fetal autopsy has been collected prospectively for each case All variates were collected in the same way over the whole time period although since 2016 saved in the Swedish Pregnancy Register. Each case was systematically reviewed by the Stockholm Stillbirth Group, consisting of obstetricians from all delivery units in Stockholm County and a senior perinatal pathologist. The cause of stillbirth was determined according to the Stockholm Stillbirth Classification [13] used since 2002. This classification defines seventeen different causes of stillbirth with clear definitions of necessary parameters for inclusion for each defined cause, Table 1. The primary and secondary cause of death as well as the degree of certainty of the cause according to classification criteria was decided by consensus during regular audits in the Stockholm Stillbirth Group [13]. The Stockholm Stillbirth Group has meeting for audit discussion approximately five times a year.

All singleton cases of stillbirth diagnosed at GW 37+0 or later were included in this study. Stillbirths diagnosed between GW 37+0 and 40+6 were compared with stillbirths diagnosed from GW 41+0 and onwards. Pregnancy and maternal characteristics have been prospectively gathered from antenatal records for all cases. Stillbirth was defined as fetal death from GW 22 +0 according to the WHO and International Classification of Diseases (ICD-10) definition [14]. In Sweden the definition of stillbirth changed from fetal death from GW 28+0 to fetal death from GW 22+0 in 2008, but the Stockholm Stillbirth Group has been using the WHO and ICD-10 definition since 1998.

Gestational age was based on the routine screening ultrasound at GW 18-20. From 2015 gestational age was based on ultrasound in GW 11+0-13+6 if the woman had an ultrasound for nuchal translucency and biparietal diameter was over 21 mm. Body mass index was based on self-reported height and measured weight at the first antenatal visit during first trimester. Parity was handled as a categorical variable, nulliparous yes/no. Maternal age was handled as a

**Table 1. Classification of stillbirth according to the Stockholm Stillbirth Classification.**

| Cause of stillbirth |
| --- |
| Malformations and chromosomal abnormalities |
| Infection |
| Immunization |
| Feto-maternal transfusion |
| Twin-to-twin transfusion syndrome |
| Birth asphyxia |
| Intrauterine growth restriction/placental insufficiency |
| Umbilical cord complication |
| Placental abruptio |
| Preeclampsia |
| Diabetes mellitus |
| Intrahepatic cholestasis of pregnancy |
| Uterine complication |
| Coagulation disorders |
| Other causes related to stillbirth |
| Unknown |
| Unexplained |

Cause of stillbirth according to the Stockholm Stillbirth Classification.

continuous variable. Maternal country of birth was divided in six different regions of the world, i.e. Sweden, Europe/Australia/USA, Middle East, Asia, South America and Africa South of Sahara. All variables were collected from the antenatal medical records or the records from the delivery ward.

In 2005 the recommended time for induction due to postterm pregnancy was changed from GW 43+0 to 42+0 in all delivery units in Stockholm County. In 2014 the surveillance from GW 41+0 changed in Stockholm [8] and all women were opted for an ultrasound at GW 41+0 with the aim to identify fetuses small for gestational age and to detect oligohydramnios. Amniotic fluid was measured as single deepest pocket, > 20 mm was considered normal [15]. The fetal abdominal diameter was measured and was considered normal if $\geq$110 mm. When the abdominal diameter was < 110 mm fetal weight was estimated [16]. Women with fetuses with estimated weight $\leq 10^{th}$ percentile were treated according to local guidelines with assessment of pulsatility index in the umbilical cord and the uterine arteries. Women with fetuses with estimated weight <- 2 standard deviations according to gestational age [17] or with abnormal Doppler findings were treated according to local guidelines and induction of labor was considered. The time periods 1998-2004, 2005-2013 and 2014-2018 were compared because of the change in recommendations in 2005 and 2014, mentioned above.

## Statistics

Frequencies of maternal and pregnancy characteristics were compared between stillbirths diagnosed between GW 37+0 and 40+6 and stillbirths diagnosed from GW 41+0 and onwards. Maternal and pregnancy variates measured on a continuous scale were presented as means and standard deviations (SD) and categorical variates as numbers and proportions. Comparisons between the continuous variates were done with Wilcoxon ranksum test and with chi-square test for the categorical variates, comparing proportions.

The number of stillbirths diagnosed from GW 41+0 and onwards were counted per year of birth and in addition as a proportion of births per year.

The proportions of different causes of stillbirth and the degree of certainty according to the Stockholm Stillbirth Classification [13] were compared between stillbirths diagnosed between GW 37+0 and 40+6 and stillbirths diagnosed from GW 41+0 and onwards, with chi-square test.

The total incidence of stillbirth in Stockholm County was calculated as well as the incidence of stillbirths diagnosed from GW 41+0 and onwards. The total incidence of stillbirth as well as the incidence of stillbirth from GW 41+0 were compared between three chosen time periods (1998-2004, 2005-2013 and 2014-2018), with chi-square test, testing the difference in proportions between these three time periods. The incidence of stillbirth from GW 41+0 per 1000 women still pregnant at GW 41+0 was also compared between the same time periods. These time periods were chosen because time of induction due to prolonged pregnancy was changed in 2005 and surveillance from GW 41+0 were standardized and changed from 2014. All statistical analyses were done using R cran.

**Ethical approval** for this study was obtained from the Regional Research Ethics Committee at Karolinska Institute in Stockholm, Sweden Dnr 2020-01855. Data were fully anonymized before access. The Ethics Committee did not require written informed consent. The Ethics Committee prohibit data to be publicly available. However, data will be shared after an approval from the Regional Research Ethics Committee.

## Results

The total number of births in the County has increased from 18 689 in 1998 to 28 672 in 2018. The number of women giving birth at GW 41+0 or later has increased from 4992 in year 1998

**Table 2. Maternal and fetal characteristics comparing term stillbirths at GW 37+0-40+6 with term stillbirths at GW 41+0 and onwards.**

| Maternal and fetal characteristics | Term Stillbirth before GW 41+0 n = 605 | Stillbirth from GW 41+0 n = 157 | P-value |
|---|---|---|---|
| Maternal age, years | 31.81 (SD 5.22) | 32.14 (SD 5.14) | 0.478 |
| Maternal age >35 (n, %) | 181 (29.92%) | 58 (36.94%) | 0.111 |
| BMI, kg/m2 | 25.21 (SD 4.83) | 25.54 (SD 5.1) | 0.491 |
| Nullipara (n, %) | 217 (41.57%) | 73 (51.77%) | 0.038 |
| Born in Sweden (n, %) | 255 (61.74%) | 65 (61.9%) | 1 |
| Born in Africa (n, %) | 27 (6.54%) | 13 (12.38%) | 0.072 |
| Born in Middel East (n, %) | 65 (15.74%) | 13 (12.38%) | 0.48 |
| Born in South America (n, %) | 4 (0.97%) | 3 (2.86%) | 0.306 |
| Born in Asia (n, %) | 23 (5.57%) | 4 (3.81%) | 0.632 |
| Born in Euroupe/USA/Australia (n, %) | 24 (5.81%) | 5 (4.76%) | 0.857 |
| Smoking (n, %) | 23 (4.57%) | 4 (2.78%) | 0.476 |
| Assisted conseption (n, %) | 21 (3.48%) | 5 (3.18%) | 1 |
| Birthweight ≤10e percentilen (n, %) | 180 (30.35%) | 51 (32.9%) | 0.607 |
| Birthweight <-2 SD (n, %) | 100 (16.86%) | 26 (16.77%) | 1 |

Stillbirths from 1998-2018, diagnosed GW 37+0-40+6 compared to stillbirths diagnosed from GW 41+0. BMI: body mass index; SGA: small for gestational age; LGA large for gestational age. Data are presented as mean and standard deviation (SD) or n (%).

to 7015 in year 2018. During the period there were 605 singleton stillbirths diagnosed between GW 37+0 and 40+6 and 157 singleton stillbirths diagnosed from GW 41+0 and onwards. Maternal and fetal characteristics for these term stillbirths are described in Table 2. There was a significant higher proportion of nullipara among stillbirths diagnosed from GW 41+0 compared to stillbirths diagnosed between GW 37+0 and 40+6, Table 2. However, this difference was not significant when analyzing the time periods separately, S1A–S1C Table and it was not significant when the proportion of nullipara women still pregnant at GW 41+0 was taken into account.

The total incidence of stillbirth in the Stockholm County was 4.6/1000 births between 1998 and 2004, 3.9/1000 births between 2005 and 2013 and 3.4/1000 births between 2014 and 2018. The incidence changed significantly, p-value < 0.001 (Fig 1). The number of stillbirths diagnosed from GW 41+0 and onwards per year has decreased during the two past decades (Figs 1 and 2). Between 1998 and 2004 the incidence was 0.5/1000 births, between 2005 and 2013 0.3/1000 births and between 2014 and 2018 0.15/1000 births, p-value < 0.001 (Fig 2). The incidence of stillbirth from GW 41+0 has also significantly decreased between the three time periods when only including women still pregnant at GW 41+0 (1.8/1000, 1.1/1000 and 0.5/1000 respectively, p-value <0.001, Fig 2).

Main causes of stillbirth are described in Table 3, comparing causes of stillbirth between GW 37+0 and 40+6 with those diagnosed from GW 41+0. There was a significant higher proportion of infections as the cause of stillbirth diagnosed from GW 41+0 and onwards compared to those diagnosed between GW 37+0 and GW 40+6. This difference was significant during the first two time periods however, not significant during the last time period (S2A–S2C Table).

During the whole period there was an increased proportion of infants small for gestational age among all term stillbirths (Table 2), also when analyzing the time periods separately there was an increased proportion of small for gestational age fetuses (S1A–S1C Table). Four fetuses small for gestational age, defined as < -2 standard deviations, were stillborn between 2014 and 2018 (S1C Table). Two of them would not have been helped by induction at GW41+0 (one

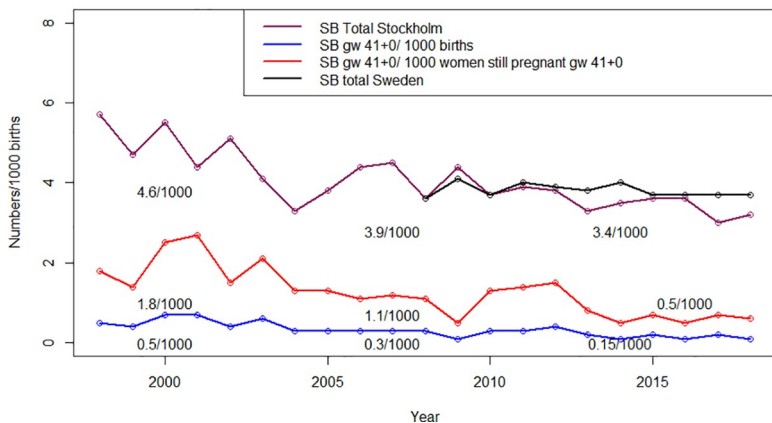

**Fig 1. Total incidence of stillbirth in Stockholm county (violet).** Incidence of stillbirth in Sweden (black). Incidence of stillbirth among women still pregnant at GW 41+0 number/1000 pregnancies (red) and incidence of stillbirth diagnosed from GW 41+0/ 1000 births (blue). The decreased incidences, violet, red and blue were all significant with p-value <0.001. The incidence of stillbirth in Stockholm County has decreased comparing three time periods (1998-2004, 2005-2013 and 2014-2018).

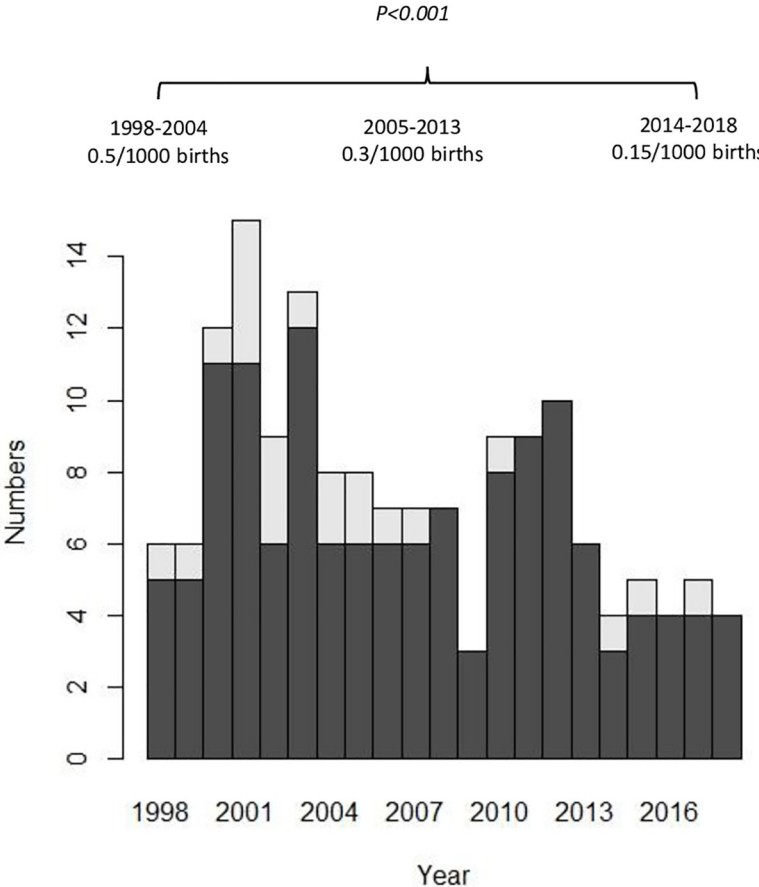

**Fig 2. Number of stillbirth cases diagnosed from GW 41+0.** Incidences compared between three time periods (1998-2004, 2005-2013 and 2014-2018). Dark grey indicates cases in GW 41 and light grey indicates cases in GW 42.

**Table 3. Main cause of stillbirth according to the Stockholm Stillbirth Classification comparing term stillbirths at GW 37+0 -40+6 with term stillbirths at GW 41+0 and onwards.**

| Main Cause of Stillbirth | Term Stillbirth before GW 41+0 n = 605 | Stillbirth from GW 41+0 n = 157 | P-value |
|---|---|---|---|
| Malformation/chromosomal abnormalities (n, %) | 45 (7.77%) | 7 (4.6%) | 0.226 |
| Infection (n, %) | 122 (21.07%) | 65 (42.2%) | <0.001 |
| Feto-maternal transfusion (n, %) | 20 (3.45%) | 1 (0.7%) | 0.113 |
| Placental insufficiency/IUGR (n, %) | 169 (29.19%) | 39 (25.3%) | 0.398 |
| Umbilical cord complications (n, %) | 55 (9.5%) | 7 (4.6%) | 0.072 |
| Placental abruptio (n, %) | 43 (7.43%) | 7 (4.6%) | 0.28 |
| Preeclampsia (n, %) | 8 (1.38%) | 2 (1.3%) | 1 |
| Diabetes mellitus (n, %) | 11 (1.9%) | 1 (0.7%) | 0.466 |
| Intrahepatic cholestasis (n, %) | 4 (0.69%) | 0 (0%) | 0.675 |
| Coagulation disorder (n, %) | 1 (0.17%) | 2 (1.3%) | 0.217 |
| Other causes related to stillbirth (n, %) | 12 (2.07%) | 3 (2.0%) | 1 |
| Cause of stillbirth un-known (n, %) | 80 (13.82%) | 18 (11.7%) | 0.578 |

Cause of stillbirth among stillbirths diagnosed GW 37+0-40+6 compared to stillbirths diagnosed from GW 41+0 from 1998-2018. Data are presented as n (%).

had a known lethal fetal abnormality; one was found dead at the opted ultrasound at 41+0 GW).

## Discussion

The incidence of stillbirth diagnosed from GW 41+0 and onwards has decreased from 0.5 to 0.15 /1000 births in Stockholm County between 1998 and 2018. Among women still pregnant at GW 41+0 the incidence of stillbirth has decreased from 1.8/1000 pregnancies to 0.5/1000 pregnancies. The total incidence of stillbirths from GW 22+0 in Stockholm County during the same time period has also decreased. However the incidence of stillbirth in all of Sweden has remained unchanged during this time period [3, 18]. Among stillbirths diagnosed from GW 41+0 and onwards there was an increased proportion of nullipara. However, comparing the proportion of nulliparas still pregnant at GW 41+0 and the proportion of nulliparas with still-birth diagnosed from GW 41+0 the difference was not significant. There was an increased pro-portion of stillbirths caused by infections among those diagnosed from GW 41+0 compared to those diagnosed between GW 37+0 and 40+6. The altered surveillance policy of post term pregnancies after 2014 has not altered the proportion of small for gestational age infants among stillbirths compared to the period 1998-2013. Table 2 and S1A–S1C Table.

The strength of the present study is the large number of consecutive stillbirths over the period. All cases in the Stockholm County are included, scrutinized and diagnosed according to the same investigation protocol and classification. There are no cases of late abortions among the stillbirth cases. The limitation of the present study includes the observational design with its difficulties to pinpoint reasons behind the decreasing incidence of stillbirths diagnosed from GW 41+0 as well as reasons behind the overall decreased incidence of stillbirths in Stockholm County.

The decreasing incidence of stillbirth diagnosed from GW $\geq$ 41+0 may be a result of changes in policy. In 2005 the time of induction of labor due to postterm pregnancy was changed from GW 43+0 to 42+0 in the County. Neonatal mortality and morbidity decreased in Stockholm County when comparing the periods 2000-2004 and 2005-2007 [7]. In 2014 an additional ultrasound scan at GW 41+0 was introduced in all delivery units in Stockholm County with the aim to identify small for gestational age fetuses and detect oligohydramnios

giving the possibility to induce labor because of these ominous signs [8]. However, the proportion of stillborn fetuses with birth weight <-2 standard deviations did not significantly change when comparing stillbirths before 2014 and stillbirths between 2014 and 2018. This is in line with results from a large randomized study showing that repeated ultrasound examinations during the third trimester resulted in a slightly increased proportion of known SGA fetuses at birth without any difference in neonatal outcome [19].

The incidence of stillbirth varies between European countries and there is a slightly decreasing trend at all gestational ages in Europe. This could suggest multifactorial causes behind the decrease [20]. Probably that is the case even in Stockholm County. The decreased incidence of stillbirths diagnosed ≥ 41+0 may be caused by changes in policy as described above together with causes affecting the overall stillbirth incidence. General obstetric care has evolved which could also affect the incidence [20]. Prenatal screening possibilities have evolved with a larger possibility to find fetal anomalies earlier in pregnancy leading to the possibility to terminate pregnancies at an earlier stage [21]. Doppler ultrasound has given possibility to surveil high-risk pregnancies more intricately. Third trimester ultrasound may have the possibility to identify a larger proportion of small for gestational age fetuses [19, 22, 23]. However, a third trimester ultrasound in a low risk population has not been shown to affect outcome for the infants [19, 22, 23]. Guidelines for the diagnosis and definition of gestational diabetes as well as preeclampsia and the recommended time of induction of labor due to preeclampsia have changed during the period [24–26]. During the past two decades the obstetric population has also changed which could affect the incidence of stillbirth. The prevalence of smoking during pregnancy has decreased [27], the prevalence of obesity has increased [18, 28] as well as maternal age [18]. Foreign-born women, especially those born in sub-Saharan Africa and the Middle East, are overrepresented in the group of women affected by stillbirth [29], Sweden has had an increasing group of women born outside Europe. During the first time period there was a very high proportion of women with unknown country of origin. The proportion of women with unknown country of origin is substantial also during the two later time periods, however much lower than during the first time period. Hence, un-known country of origin is not a substantial explanation of the decreasing number of women with stillbirth born in Sweden comparing the two later time periods. Changes in the obstetric population could affect the incidence of stillbirth.

There was an increased incidence of infection as primary cause of stillbirth among stillbirths diagnosed from GW 41+0. A histological diagnosis of chorioamnionitis is more common in term pregnancies than preterm pregnancies [30]. However, to be able to classify infection as a definite or probable cause of stillbirth, additional findings are needed according to the Stockholm Stillbirth Classification [13] such as vasculitis in the placenta or umbilical cord or funicitis or positive cultures from amnion fluid or fetal or maternal blood. To what degree the histological signs of chorioamnionitis are merely signs of infection with potential to harm the fetus or signs of inflammation associated with the delivery or the late GW in general is still unclear.

A decreasing incidence of stillbirth is observed in Stockholm County. However, to what degree induction of labor at GW 41+0 will contribute to the continuing declining incidence still is open for debate. Probably the answer is more complex, and induction of labor at an earlier gestational age will not solely solve the problem. Other parameters as body mass index, maternal country of birth, educational level and other socioeconomic factors are risk factors but the mechanisms behind these associations are largely unknown. More knowledge is still needed to make surveillance more focused on relevant risk factors.

## Conclusion

There is a decreasing incidence of stillbirths diagnosed from GW 41+0 onwards in Stockholm County during the period 1998-2018. There is in addition, an overall decreasing incidence of stillbirths in Stockholm County during the same period. Infection was more common as a cause of death among stillbirths diagnosed from GW 41+0.

Due to the study design the underlying causes behind this decreasing incidence of stillbirth cannot be clearly identified in this study. The prevalence of risk factors in the pregnant population have changed over time and so has monitoring and mode of treatment. Further studies are needed to identify causes behind the reduced numbers of stillbirths.

## Supporting information

**S1 Table. a.** Maternal and fetal characteristics comparing term stillbirths at GW 37+0-40+6 with term stillbirths at GW 41+0 and onwards between 1998-2004. **b.** Maternal and fetal characteristics comparing term stillbirths at GW 37+0-40+6 with term stillbirths at GW 41+0 and onwards between 2005-2013. **c.** Maternal and fetal characteristics comparing term stillbirths at GW 37+0-40+6 with term stillbirths at GW 41+0 and onwards between 2014-2018.
(DOCX)

**S2 Table. a.** Main cause of stillbirth according to the Stockholm Stillbirth Classification comparing term stillbirths at GW 37+0 -40+6 with term stillbirths at GW 41+0 and onwards between 1998-2004. **b.** Main cause of stillbirth according to the Stockholm Stillbirth Classification comparing term stillbirths at GW 37+0 -40+6 with term stillbirths at GW 41+0 and onwards between 2005-2013. **c.** Main cause of stillbirth according to the Stockholm Stillbirth Classification comparing term stillbirths at GW 37+0 -40+6 with term stillbirths at GW 41+0 and onwards between 2014-2018.
(DOCX)

## Acknowledgments

We would like to thank the Stockholm Stillbirth Group for a fantastic and assiduous work with collection and classification of all cases of stillbirth for 20 years.

## Author Contributions

**Conceptualization:** Hanna Åmark, Christina Pilo, Ingela Hulthén Varli.

**Formal analysis:** Hanna Åmark.

**Methodology:** Hanna Åmark, Christina Pilo, Ingela Hulthén Varli.

**Writing – original draft:** Hanna Åmark, Christina Pilo, Ingela Hulthén Varli.

**Writing – review & editing:** Hanna Åmark, Christina Pilo, Ingela Hulthén Varli.

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
