## [Decision Letter · Decision Letter 0]

3 Mar 2021

PONE-D-21-01718

Stillbirth in Stockholm during a 20-year period, incidence and causes with focus on term and late term gestations

PLOS ONE

Dear Dr. Åmark,

Thank you for submitting your manuscript to PLOS ONE. After careful consideration, we feel that it has merit but does not fully meet PLOS ONE’s publication criteria as it currently stands. Therefore, we invite you to submit a revised version of the manuscript that addresses the points raised during the review process.

We look forward to receiving your revised manuscript.

Kind regards,

David Desseauve, MD, MPH, PhD

Academic Editor

PLOS ONE

Journal Requirements:

2)  Thank you for stating the following in the Acknowledgments Section of your manuscript:

[We would also like to thank Spädbarnsfonden and KI Research Foundation for supporting this

project.]

 [This study was supported by Spädbarnsfonden.]

3)  We note that you have indicated that data from this study are available upon request. PLOS only allows data to be available upon request if there are legal or ethical restrictions on sharing data publicly. For information on unacceptable data access restrictions, please see http://journals.plos.org/plosone/s/data-availability#loc-unacceptable-data-access-restrictions.

Reviewers' comments:

Reviewer's Responses to Questions

**Comments to the Author**

1. Is the manuscript technically sound, and do the data support the conclusions?

Reviewer #1: Partly

Reviewer #2: Yes

Reviewer #3: Yes

Reviewer #4: Yes

2. Has the statistical analysis been performed appropriately and rigorously? 

Reviewer #1: Yes

Reviewer #2: Yes

Reviewer #3: Yes

Reviewer #4: Yes

3. Have the authors made all data underlying the findings in their manuscript fully available?

Reviewer #1: No

Reviewer #2: No

Reviewer #3: Yes

Reviewer #4: No

4. Is the manuscript presented in an intelligible fashion and written in standard English?

Reviewer #1: Yes

Reviewer #2: Yes

Reviewer #3: Yes

Reviewer #4: Yes

5. Review Comments to the Author

Reviewer #1: The topic is very interesting and it is quite orginal to focus on late stillbirth.

The manuscript is exclusively discribing late stillbirth so the title is ambiguous according to me, I would have chosen "Stillbirth in term or late gestation in Stockholm during a 20-year period..."

Discribing causes of stillbirth according to the gestational age gives relevant informations. But the comparison of maternal caracteristics between pregnancies before and after 41WG is obvioulsy related to the factors associated to late pregnancies such as primiparity. The authors discuss the fact that ultrasound performed at 41WG did not lead to reduce the rate of stillbirth due to SGA, though the performance of late ultrasound should here be discussed.

Comparison of very small group of patients in tables 2a and 3a can not be conclusive (n=22).

figure 2 why did the authors excluded late stillbirth while that is the subject of the mansucript, why not showing the rate /1000 birth of late stillbirth?

Reviewer #2: This paper from Stockholm Stillbirth Database reports stillbirth from the Stockholm county between 1998 and 2018. Not many countries have access to this kind of data.

Although purely descriptive as mentioned by authors in the discussion, and honestly not including very new data, this paper is arriving at the right time.

Since the publication of Grobman et al. study (Grobman WA, Rice MM, Reddy UM, Tita ATN, Silver RM, Mallett G, Hill K, Thom EA, El-Sayed YY, Perez-Delboy A, Rouse DJ, Saade GR, Boggess KA, Chauhan SP, Iams JD, Chien EK, Casey BM, Gibbs RS, Srinivas SK, Swamy GK, Simhan HN, Macones GA; Eunice Kennedy Shriver National Institute of Child Health and Human Development Maternal–Fetal Medicine Units Network. Labor Induction versus Expectant Management in Low-Risk Nulliparous Women. N Engl J Med. 2018;379(6):513-523) some teams around the world are thinking of inducing women at 39 weeks.

I would recommend to authors to try to analyze their data in order to look at data before and after 39 weeks.

I also would retrieve the cases with fetal malformations, TTS and immunization as independent factors and not considered as low risk pregnancies

Reviewer #3: I would like to thank the authors for this article, which studies the incidence of term and later stillbirths on a large period ( 1998-2018) in Stockholm County. The authors divided this period in three unequal laps of time ( 1998-200,/2005-2013, 2014-2018) due to changes in the management of late term pregnancies. More, the authors provided information on this etiology of each stillbirths.

However, it is difficult to have a correct overview of how the changes in the management of late term/ prolonged pregnancy have influenced the rate of stillbirths in this population?

Although the different curves are enough to understand the decrease in the incidence of stillbirths, Comparison groups were inadequate to understand if the population with stillbirths was also associated with different characteristics.

That being said, even though the overall is well written, there are sections that would benefit from rewriting, as the authors tend to lose sight of their objectives.

Here are some remarks I would like to share with the authors:

For the abstract:

The authors should state more clearly the objective of the study. ( to evaluate the incidence…)

The term of primipara should be avoided and nulliparous should be preferred. This modification should be done all along the manuscript.

Introduction :

There is a mistake in the incidence reported of stillbirths?. (wrong reference?)Reference 3 not available.

The third paragraph dealing with the etiologies of stillbirth is too long, and some sentences are inappropriate with the subject (etiologies of preterm births). The authors should focus on the etiologies of stillbirths in term and late-term pregnancies.

Material and methods :

The authors referred to two registers: one is the Stockholm Stillbirth database and the second one the Swedish Pregnancy Register. Did these two registers collected the same data? How the variables were recorded?

During this period of 20 years, was the cause of stillbirths always determined by two physicians ? (obstetrician and a perinatal pathologist)

The Stockholm Stillbirth Classification should be more detailed: how was it validated? Is there a good agreement between physicians in the determination of one etiology of the stillbirth. What is the proportion of stillbirths which remain unexplained? This information is important to validate the results presented afterwards. Since when this classification was used? This should be discussed.

What was the audit? how often is it?

I am doubtful with the description of the group 37+0 -40+6 weeks? To my point of view, the authors evaluate how modifications of late term/ prolonged pregnancies have influenced the incidence of stillbirths in this population. Therefore, description of this latest group ( ≥41 +0 ) is enough. If no, it should be interesting to provide information on the overall incidence of stillbirths.

The dating of the beginning of the pregnancy seemed to be based on routine ultrasound at 18-20 weeks. Was it the case during all the study? What about the first trimester scan? This should be discussed, as we could imagine that a better determination of the beginning of the pregnancy would have allowed to decrease the probability of a real prolonged pregnancy?

Maternal age seemed to be unchanged when it is described with means. However, it would be appreciated if maternal age was handled also a categorial variable ( to see if the proportion of women aged more than≥35 or 40 years was modified.)

If there is no information on the country of patients who were born out of Sweden , this variable should be removed.

Can the authors quote the reference which say that oligoamnios was related to as single deepest pool ≤20 mm? Same for AD≤110mm?

Results:

Were there 6 maternity units during all the period? If no , please remove this sentence.

The results presented referred to the description of maternal characteristics in the overall population study (1998-2018) and those related to women of the last period (2014-2018). There is no description of maternal characteristics of the two previous groups?

How the infectious cause was retained? What were the elements which were requested to keep this etiology?

A comparison of maternal characteristics for each period of time would be more adequate, than comparison between stillbirths occurring before and after 41 weeks of gestation.

The primary result (incidence of stillbirth) should be presented before the causes of stillbirth.

Table 3 reports only 12 different causes of stillbirths, whereas the classification used described 17 causes (table 1). How many stillbirths were unexplained between the three periods?

Discussion :

The discussion is difficult to read, as the ideas follow without order or hamony.

Legend 2 : evolution of stillbirths in Sweden should be removed.

The discussion should be

.

Reviewer #4: Your article titled Stillbirth in Stockholm during a 20-year period, incidence and causes with focus on term and late term gestations is submitted for publication in Plos One. My first remark should encourage you to explain to the reader why Stockholm is a good place to study the future of stillbirths.

-1- Do you consider Stockholm's rates to be among the lowest in the world? Is the question of continuing a long-term gestation beyond 41 GW or of interrupting labor at the 41th GW a priority question that arises or will arise in all countries? In other words, does the situation in Sweden, and in particular Stockholm, serve as an example for other countries?

We want to believe it, but the increase in the average age of women at the birth of their children, the heavy smoking during pregnancy, and the fight against social discrimination are also priorities for the countries even among the richest of the planet?

It seems to me that you should indeed give more arguments because other very recent publications relate exactly to the same subject (PLOS medicine 2020) and we would like your article to be able to both distinguish itself from it and also demonstrate that the subject is important for other countries, see reference (* from the Euro-Peristat Project) for example, but you mentioned an earlier reference [20].

-2- You seem to think that your statistical measurements of both gestational age and the number of stillbirths have not been affected by errors, of course, but when we know the statistical difficulties in certain countries in distinguishing abortions for therapeutic causes from stillbirths, the international reader would like to have a sentence to be enlightened on this point.

You could argue that to avoid the statistical problem of termination of pregnancy, you chose to primarily study long-term pregnancy of 40 GW to 42+ because the effect of termination of pregnancy is large only at short gestation times. However, this should affect the statistical measure of Sweden's stillbirth rate at the standard 36 GW threshold and you could mention this point.

-3- Your article, like other similar articles, does not frankly conclude on the benefit of inducing late pregnancy and your discussion is very interesting. But one wonders what would you need to be able to conclude and not just say that the woman must be informed of the risks incurred in each two cases of either termination or continuation of the pregnancy. And it seems from what you write that you need more statistical power, and you wonder why you do expand to the entire Sweden, or even why you do not prefer to do individual participant data meta-analyzes as of other colleagues. One of your main reference concerns a paper published in January 2019 in BMJ, but a paper published in December 2020 in Plos medicine should be mentioned.

-4- It is difficult to get a figure related to the title of your article: 20 years period of stillbirth in Stockholm? Figure 2, should be the place but there is no definition of stillbirths (22GW, 37GW ?). Stagnation of Sweden a whole since 10 years is not discussed.

-5- An important point that you do not discuss concerns the high proportion of women born outside Sweden (48% table 2) and how it is related to stillbirth rate.

-6- How a relatively higher proportion of primiparus women (46%) and its rapid fall between 2010 and 2015 (43.1) [3 figure C9.2] at least will affect the results. The prevalence of smoking during pregnancy has decreased [28] seems to be true concerning Europe according to [3 Table R8.1] but discrepancies between countries are huge.

(*) Blondel B, Cuttini M, Hindori-Mohangoo AD, et al. How do late terminations of pregnancy affect comparisons of stillbirth rates in Europe? Analyses of aggregated routine data from the Euro-Peristat Project. BJOG. 2018;125(2):226-34. doi: 10.1111/1471-0528.14767.

6. PLOS authors have the option to publish the peer review history of their article (what does this mean?). If published, this will include your full peer review and any attached files.

Reviewer #1: No

Reviewer #2: No

Reviewer #3: No

Reviewer #4: No

---

## [Author Response · Author response to Decision Letter 0]

27 Apr 2021

Reviewer #1: The topic is very interesting and it is quite orginal to focus on late stillbirth.

The manuscript is exclusively discribing late stillbirth so the title is ambiguous according to me, I would have chosen "Stillbirth in term or late gestation in Stockholm during a 20-year period..."

Discribing causes of stillbirth according to the gestational age gives relevant informations. But the comparison of maternal caracteristics between pregnancies before and after 41WG is obvioulsy related to the factors associated to late pregnancies such as primiparity. The authors discuss the fact that ultrasound performed at 41WG did not lead to reduce the rate of stillbirth due to SGA, though the performance of late ultrasound should here be discussed.

We thank the reviewer for this comment. We have compared the proportion of nullipara still pregnant at gw 41+0 and the proportion of nullipara with sb from 41+0 and there is no significant difference. We have changed that throughout the manuscript. We agree that ultrasound examinations have limitations. In the discussion we have included: “This is in line with results from a large randomized study showing that repeated ultrasound examinations during the third trimester resulted in a slightly increased proportion of known SGA fetuses at birth without any difference in neonatal outcome [1]. “ 

Comparison of very small group of patients in tables 2a and 3a can not be conclusive (n=22).

We have put table 1a and 2a in supplementary material. We agree that there are few cases in that group. 

figure 2 why did the authors excluded late stillbirth while that is the subject of the mansucript, why not showing the rate /1000 birth of late stillbirth?

We have included rate of stillbirth gw 41+0/1000 births and rate of stillbirth gw 41+0/1000 women still pregnancies still pregnant in that gestational week. 

Reviewer #2: This paper from Stockholm Stillbirth Database reports stillbirth from the Stockholm county between 1998 and 2018. Not many countries have access to this kind of data.

Although purely descriptive as mentioned by authors in the discussion, and honestly not including very new data, this paper is arriving at the right time.

Since the publication of Grobman et al. study (Grobman WA, Rice MM, Reddy UM, Tita ATN, Silver RM, Mallett G, Hill K, Thom EA, El-Sayed YY, Perez-Delboy A, Rouse DJ, Saade GR, Boggess KA, Chauhan SP, Iams JD, Chien EK, Casey BM, Gibbs RS, Srinivas SK, Swamy GK, Simhan HN, Macones GA; Eunice Kennedy Shriver National Institute of Child Health and Human Development Maternal–Fetal Medicine Units Network. Labor Induction versus Expectant Management in Low-Risk Nulliparous Women. N Engl J Med. 2018;379(6):513-523) some teams around the world are thinking of inducing women at 39 weeks.

I would recommend to authors to try to analyze their data in order to look at data before and after 39 weeks.

We thank the reviewer for this comment, it is indeed an interesting question. However, it was not the scoop of this paper and this question will have to wait till another occasion. 

I also would retrieve the cases with fetal malformations, TTS and immunization as independent factors and not considered as low risk pregnancies.

We are not sure about what the reviewer wants. There are some cases with malformations known or un-known. There are no cases caused by immunization since they occur in earlier gestational weeks. Twin pregnancies are excluded. 

Reviewer #3: I would like to thank the authors for this article, which studies the incidence of term and later stillbirths on a large period ( 1998-2018) in Stockholm County. The authors divided this period in three unequal laps of time ( 1998-200,/2005-2013, 2014-2018) due to changes in the management of late term pregnancies. More, the authors provided information on this etiology of each stillbirths.

However, it is difficult to have a correct overview of how the changes in the management of late term/ prolonged pregnancy have influenced the rate of stillbirths in this population?

We thank the reviewer for this comment. We agree that it is not only difficult but in fact impossible to know how the changes in management have influences the stillbirth rate. It may be other differences, perhaps more general and probably multifactorial that have influenced the stillbirth rate. Since this is an observational study without the possibility to control for all other differences between the different time periods, we cannot answer why the rates have decreased. That is a limitation, however stillbirth is a rare outcome with methodological difficulties associated to that. One thing we can conclude is that the number of stillbirths has decreased as well as the proportion of stillbirth, however over all the years the proportion of stillborn SGA fetuses are the same over all time periods. We have managed to decrease the numbers however it seems we have not lowered the risk for the SGA fetuses. 

Although the different curves are enough to understand the decrease in the incidence of stillbirths, Comparison groups were inadequate to understand if the population with stillbirths was also associated with different characteristics.

That being said, even though the overall is well written, there are sections that would benefit from rewriting, as the authors tend to lose sight of their objectives.

Here are some remarks I would like to share with the authors:

For the abstract:

The authors should state more clearly the objective of the study. ( to evaluate the incidence…) The term of primipara should be avoided and nulliparous should be preferred. This modification should be done all along the manuscript.

We thank the reviewer for these comments and have changed primipara to nulliparous. We have also made the objective clearer and now write:

“A recent report has shown Stockholm to have a lower incidence of stillbirth at term than most of the country”

Introduction :

There is a mistake in the incidence reported of stillbirths?. (wrong reference?)Reference 3 not available.

We thank the reviewer for this comment and have changed reference 3, we thank the reviewer for the comment on incidence, it was too many zeros, we have changed that. 

The third paragraph dealing with the etiologies of stillbirth is too long, and some sentences are inappropriate with the subject (etiologies of preterm births). The authors should focus on the etiologies of stillbirths in term and late-term pregnancies.

We have rewritten paragraph 3 and we now write “A recent study comparing induction of labor at GW 41+0 to expectance and induction latest at GW 42+0 showed a lower risk of stillbirth in the former group [2]. These results and varying management policies between countries regarding postterm pregnancies have actualized the discussion regarding optimal time of induction of labor due to prolonged pregnancy. Fetal abnormalities, placental insufficiency, fetal growth restriction and infections are all common causes of stillbirth as is placental abruptio and umbilical cord complications [3, 4]. Main causes of stillbirth vary with gestational age [3, 5]. Pregnancies passing gestational week 41+0 are, in general, women without risk factors and without known pregnancy complications, hence to a large extent healthy women with healthy fetuses.”

Material and methods :

The authors referred to two registers: one is the Stockholm Stillbirth database and the second one the Swedish Pregnancy Register. Did these two registers collected the same data? How the variables were recorded?

We thank the reviewer for this comment. All variables used were the same and recorded in the same way for all years and for the two registers. We have made it more clear how variables were recorded. 

During this period of 20 years, was the cause of stillbirths always determined by two physicians ? (obstetrician and a perinatal pathologist)

During the period of 20 years the cause of stillbirth was always determined by one perinatal pathologist and one obstetrician from each delivery ward in the county, a group of 7-10 persons, based on the classification, to secure that it was done the same way all the time. The perinatal pathologist and some of the obstetricians have been the same individuals for the whole time. 

The Stockholm Stillbirth Classification should be more detailed: how was it validated? Is there a good agreement between physicians in the determination of one etiology of the stillbirth. What is the proportion of stillbirths which remain unexplained? This information is important to validate the results presented afterwards. Since when this classification was used? This should be discussed. What was the audit? how often is it?

We thank the reviewer for this comment. The Stockholm classification is used since 2002., Before 2002 we used a modified classification described by the Stockholm stillbirth group in: (Petersson K, Bremme K, Bottinga R, Hofsjö A, Hulthén-Varli I, Kublickas M, et al. Diagnostic evaluation of intrauterine et al death in Stockholm 1998-99. Acta Obstet Gynecol Scand. 2002;/81:/284-92) 

The causes of stillbirth are the same in the Stockholm classification and the earlier used classification, even though the definitions of causes are more strictly defined in the Stockholm classification why it is unlikely to have any significant impact on the primary causes included in this study. The causes in the classification is based on facts from the post partal investigations and with clear information about how to judge every found sign of pathology. The classification is validated and showed a high degree of agreement (described in ref 16). We have added additional information in the methods section. There are audit meetings approximately 5 times a year. There are clear definitions of every cause of death and what is claimed for the certainty of the cause. The work and the group have been almost the same since 2002. There are between 12-13% cases which remain unexplained. 

I am doubtful with the description of the group 37+0 -40+6 weeks? To my point of view, the authors evaluate how modifications of late term/ prolonged pregnancies have influenced the incidence of stillbirths in this population. Therefore, description of this latest group ( ≥41 +0 ) is enough. If no, it should be interesting to provide information on the overall incidence of stillbirths.

We are not completely sure that we understand what the reviewer wants. We think that it is valuable to compare the late term stillbirths with another group of stillbirths and we find full term stillbirths before GW 41+0 a valuable comparison group. The overall incidence of stillbirth is provided (from GW 22+0). 

The dating of the beginning of the pregnancy seemed to be based on routine ultrasound at 18-20 weeks. Was it the case during all the study? What about the first trimester scan? This should be discussed, as we could imagine that a better determination of the beginning of the pregnancy would have allowed to decrease the probability of a real prolonged pregnancy?

We thank the reviewer for this comment. The dating was based on the routine ultrasound until 2014. From 2015 the dating was based on the first trimester ultrasound in GW 11-13 if the first trimester ultrasound was done and the biparietal diameter was >21mm. If biparietal diameter was <21 mm or if the first trimester ultrasound was not performed the dating was based on the routine ultrasound in GW 18-20. Information added in the methods section. 

Maternal age seemed to be unchanged when it is described with means. However, it would be appreciated if maternal age was handled also a categorial variable ( to see if the proportion of women aged more than≥35 or 40 years was modified.)

We thank the reviewer for this comment. We have added information on proportions of women aged >35 and >40 years. 

If there is no information on the country of patients who were born out of Sweden , this variable should be removed.

We have added more detailed information about region of maternal birth. 

Can the authors quote the reference which say that oligoamnios was related to as single deepest pool ≤20 mm? Same for AD≤110mm?

We have added references. 

Results:

Were there 6 maternity units during all the period? If no, please remove this sentence.

We have excluded that sentence. During a short time 140303 – 160531 there were 7 maternity units in Stockholm county. 

The results presented referred to the description of maternal characteristics in the overall population study (1998-2018) and those related to women of the last period (2014-2018). There is no description of maternal characteristics of the two previous groups?

We have added tables comparing all time periods in supplementary material. 

How the infectious cause was retained? What were the elements which were requested to keep this etiology?

We thank the reviewer for this comment. The diagnosis was set according to the Stockholm Stillbirth classification and for a definite diagnosis signs of fetal pneumonia or positive fetal heart blood or amnion culture and signs of placental infection was claimed. For a probable diagnosis placental signs of chorioamnionitis plus vasculitis or funicitis or signs of choriomnionitis in placenta together with clinical signs of chorioamnionitis or positive cultures. 

A comparison of maternal characteristics for each period of time would be more adequate, than comparison between stillbirths occurring before and after 41 weeks of gestation.

We have added tables with comparisons for each time period in supplementary. 

The primary result (incidence of stillbirth) should be presented before the causes of stillbirth.

We thank the reviewer for this comment, we have changed and now present the incidences first. 

Table 3 reports only 12 different causes of stillbirths, whereas the classification used described 17 causes (table 1). How many stillbirths were unexplained between the three periods?

The reported causes of death were the causes with at least one case. There were 13 % unexplained stillbirths between 1998-2004, 12.94% between 2005-2013 and 12.36% between 2014-2018.

Discussion :

The discussion is difficult to read, as the ideas follow without order or hamony.

Legend 2 : evolution of stillbirths in Sweden should be removed.

We cant find that sentence. 

The discussion should be

It seems to be missing information here. 

.

Reviewer #4: Your article titled Stillbirth in Stockholm during a 20-year period, incidence and causes with focus on term and late term gestations is submitted for publication in Plos One. My first remark should encourage you to explain to the reader why Stockholm is a good place to study the future of stillbirths.

-1- Do you consider Stockholm's rates to be among the lowest in the world? Is the question of continuing a long-term gestation beyond 41 GW or of interrupting labor at the 41th GW a priority question that arises or will arise in all countries? In other words, does the situation in Sweden, and in particular Stockholm, serve as an example for other countries?

Please see comment 3 reviewer 3. And in addition, when to induce labour is of course of relevance in Sweden as in other countries and the interest elevated in Sweden after the so called SWEPIS-study (ref 10). 

The finding in the report from the National Board of Health and Welfare, that the incidence of term stillbirth was lower in Stockholm County than most of other counties in Sweden, make Stockholm an interesting place to study term and late term stillbirths.

We want to believe it, but the increase in the average age of women at the birth of their children, the heavy smoking during pregnancy, and the fight against social discrimination are also priorities for the countries even among the richest of the planet?

It seems to me that you should indeed give more arguments because other very recent publications relate exactly to the same subject (PLOS medicine 2020) and we would like your article to be able to both distinguish itself from it and also demonstrate that the subject is important for other countries, see reference (* from the Euro-Peristat Project) for example, but you mentioned an earlier reference [20].

We thank the reviewer for an interesting comment, can you please name the reference since we have not found it. 

-2- You seem to think that your statistical measurements of both gestational age and the number of stillbirths have not been affected by errors, of course, but when we know the statistical difficulties in certain countries in distinguishing abortions for therapeutic causes from stillbirths, the international reader would like to have a sentence to be enlightened on this point.

We thank the reviewer for this important comment. We have added: “There are no cases of late abortions among the stillbirth cases. “ 

You could argue that to avoid the statistical problem of termination of pregnancy, you chose to primarily study long-term pregnancy of 40 GW to 42+ because the effect of termination of pregnancy is large only at short gestation times. However, this should affect the statistical measure of Sweden's stillbirth rate at the standard 36 GW threshold and you could mention this point.

All stillbirth cases have been scrutinized and there are no cases of late abortion among them. 

-3- Your article, like other similar articles, does not frankly conclude on the benefit of inducing late pregnancy and your discussion is very interesting. But one wonders what would you need to be able to conclude and not just say that the woman must be informed of the risks incurred in each two cases of either termination or continuation of the pregnancy. And it seems from what you write that you need more statistical power, and you wonder why you do expand to the entire Sweden, or even why you do not prefer to do individual participant data meta-analyzes as of other colleagues. One of your main reference concerns a paper published in January 2019 in BMJ, but a paper published in December 2020 in Plos medicine should be mentioned.

We thank the reviewer for this comment and are not sure about what reference from Plos Dec 2020 should be mentioned, could you please add a title? 

-4- It is difficult to get a figure related to the title of your article: 20 years period of stillbirth in Stockholm? Figure 2, should be the place but there is no definition of stillbirths (22GW, 37GW ?). Stagnation of Sweden a whole since 10 years is not discussed.

We thank the reviewer for this comment, until June 2008 the official statistic reported in Sweden was from gw 28+0. From 1 July 2008 the official statistic reported is from gw 22+0, this is now clarified in the text note under figure 2. The numbers for Stockholm county is from GW 22+0 for all three time periods. 

-5- An important point that you do not discuss concerns the high proportion of women born outside Sweden (48% table 2) and how it is related to stillbirth rate.

We thank the reviewer for this comment, and have added a sentence of this in discussion third paragraph: “Foreign-born women, especially those born in sub-Saharan Africa and the Middle East, are overrepresented in the group of women affected by stillbirth, [6] Sweden has had an increasing group of foreign-born women since many years”

-6- How a relatively higher proportion of primiparus women (46%) and its rapid fall between 2010 and 2015 (43.1) [3 figure C9.2] at least will affect the results. The prevalence of smoking during pregnancy has decreased [28] seems to be true concerning Europe according to [3 Table R8.1] but discrepancies between countries are huge.

The proportion of primiparous women has been between 40-45% in Sweden since the 70th. It is true that the proportion of smoking has decreased as well as the proportion of pregnant women with a BMI >25. 

1. Henrichs J, Verfaille V, Jellema P, Viester L, Pajkrt E, Wilschut J, et al. Effectiveness of routine third trimester ultrasonography to reduce adverse perinatal outcomes in low risk pregnancy (the IRIS study): nationwide, pragmatic, multicentre, stepped wedge cluster randomised trial. BMJ (Clinical research ed). 2019;367:l5517. Epub 2019/10/17. doi: 10.1136/bmj.l5517. PubMed PMID: 31615781; PubMed Central PMCID: PMCPMC6792062 www.icmje.org/coi_disclosure.pdf and declare: for the current study (the IRIS study), AdJ and JW received funding from the Netherlands Organisation for Health Research and Development; no financial relationships with any organisations that might have an interest in the submitted work in the previous three years; no other relationships or activities that could appear to have influenced the submitted work.

2. Wennerholm UB, Saltvedt S, Wessberg A, Alkmark M, Bergh C, Wendel SB, et al. Induction of labour at 41 weeks versus expectant management and induction of labour at 42 weeks (SWEdish Post-term Induction Study, SWEPIS): multicentre, open label, randomised, superiority trial. BMJ (Clinical research ed). 2019;367:l6131. Epub 2019/11/22. doi: 10.1136/bmj.l6131. PubMed PMID: 31748223.

3. Stormdal Bring H, Hulthen Varli IA, Kublickas M, Papadogiannakis N, Pettersson K. Causes of stillbirth at different gestational ages in singleton pregnancies. Acta obstetricia et gynecologica Scandinavica. 2014;93(1):86-92. Epub 2013/10/15. doi: 10.1111/aogs.12278. PubMed PMID: 24117104.

4. Bodnar LM, Parks WT, Perkins K, Pugh SJ, Platt RW, Feghali M, et al. Maternal prepregnancy obesity and cause-specific stillbirth. The American journal of clinical nutrition. 2015;102(4):858-64. Epub 2015/08/28. doi: 10.3945/ajcn.115.112250. PubMed PMID: 26310539; PubMed Central PMCID: PMCPMC4588742.

5. The Stillbirth Collaborative Research Network Writing Group. Causes of death among stillbirths. JAMA. 2011;306(22):2459-68. Epub 2011/12/15. doi: 306/22/2459 [pii]

10.1001/jama.2011.1823. PubMed PMID: 22166605.

6. Ekeus C, Cnattingius S, Essen B, Hjern A. Stillbirth among foreign-born women in Sweden. European journal of public health. 2011;21(6):788-92. Epub 2011/01/13. doi: 10.1093/eurpub/ckq200. PubMed PMID: 21224278.

---

## [Editor Report · Decision Letter 1]

7 May 2021

Stillbirth in term and late term gestations in Stockholm during a 20-year period, incidence and causes.

PONE-D-21-01718R1

Dear Dr. Åmark,

We’re pleased to inform you that your manuscript has been judged scientifically suitable for publication and will be formally accepted for publication once it meets all outstanding technical requirements.

Kind regards,

David Desseauve, MD, MPH, PhD

Academic Editor

PLOS ONE
---

## [Editor Report · Acceptance letter]

14 May 2021

PONE-D-21-01718R1 

Stillbirth in term and late term gestations in Stockholm during a 20-year period, incidence and causes. 

Dear Dr. Åmark:

I'm pleased to inform you that your manuscript has been deemed suitable for publication in PLOS ONE. Congratulations! Your manuscript is now with our production department. 

Kind regards, 

on behalf of

Dr. David Desseauve 

Academic Editor

PLOS ONE